# A Bayesian Framework for Modeling Confidence in Perceptual Decision Making

**Koosha Khalvati, Rajesh P. N. Rao**
Department of Computer Science and Engineering
University of Washington
Seattle, WA 98195
{koosha, rao}@cs.washington.edu

## Abstract

The degree of confidence in one's choice or decision is a critical aspect of perceptual decision making. Attempts to quantify a decision maker's confidence by measuring accuracy in a task have yielded limited success because confidence and accuracy are typically not equal. In this paper, we introduce a Bayesian framework to model confidence in perceptual decision making. We show that this model, based on partially observable Markov decision processes (POMDPs), is able to predict confidence of a decision maker based only on the data available to the experimenter. We test our model on two experiments on confidence-based decision making involving the well-known random dots motion discrimination task. In both experiments, we show that our model's predictions closely match experimental data. Additionally, our model is also consistent with other phenomena such as the hard-easy effect in perceptual decision making.

## 1  Introduction

The brain is faced with the persistent challenge of decision making under uncertainty due to noise in the sensory inputs and perceptual ambiguity . A mechanism for self-assessment of one's decisions is therefore crucial for evaluating the uncertainty in one's decisions. This kind of decision making, called perceptual decision making, and the associated self-assessment, called confidence, have received considerable attention in decision making experiments in recent years [9, 10, 12, 13]. One possible way of estimating the confidence of a decision maker is to assume that it is equal to the accuracy (or performance) on the task. However, the decision maker's belief about the chance of success and accuracy need not be equal because the decision maker may not have access to information that the experimenter has access to [3]. For example, in the well-known task of random dots motion discrimination [18], on each trial, the experimenter knows the difficulty of the task (coherence or motion strength of the dots), but not the decision maker [3, 13]. In this case, when the data is binned based on difficulty of the task, the accuracy is not equal to decision confidence. An alternate way to estimate the subject's confidence is to use auxiliary tasks such as post-decision wagering [15] or asking the decision maker to estimate confidence explicitly [12]. These methods however only provide an indirect window into the subject's confidence and are not always applicable.

In this paper, we explain how a model of decision making based on Partially Observable Decision Making Processes (POMDPs) [16, 5] can be used to estimate a decision maker's confidence based on experimental data. POMDPs provide a unifying Bayesian framework for modeling several important aspects of perceptual decision making including evidence accumulation via Bayesian updates, the role of priors, costs and rewards of actions, etc. One of the advantages of the POMDP model over the other models is that it can incorporate various types of uncertainty in computing the optimal

---

This research was supported by NSF grants EEC-1028725 and 1318733, and ONR grant N000141310817.

decision making strategy. Drift-diffusion and race models are able to handle uncertainty in proba-
bility updates [3] but not the costs and rewards of actions. Furthermore, these models originated as
descriptive models of observed data, where as the POMDP approach is fundamentally normative,
prescribing the optimal policy for any task requiring decision making under uncertainty. In addition,
the POMDP model can capture the temporal dynamics of a task. Time has been shown to play a
crucial role in decision making, especially in decision confidence [12, 4]. POMDPs have previ-
ously been used to model evidence accumulation and understand the role of priors [16, 5, 6]. To our
knowledge, this is the first time that it is being applied to model confidence and explain experimental
data on confidence-based decision making tasks.

In the following sections, we introduce some basic concepts in perceptual decision making and
show how a POMDP can model decision confidence. We then explore the model's predictions
for two well-known experiments in perceptual decision making involving confidence: (1) a fixed
duration motion discrimination task with post-decision wagering [13], and (2) a reaction-time mo-
tion discrimination task with confidence report [12]. Our results show that the predictions of the
POMDP model closely match experimental data. The model's predictions are also consistent with
the "hard-easy" phenomena in decision making, involving over-confidence in the hard trials and
under-confidence in the easy ones [7].

## 2   Accuracy, Belief and Confidence in Perceptual Decision Making

Consider perceptual decision making tasks in which the subject has to guess the hidden *state* of the
environment correctly to get a *reward*. Any guess other than the correct state usually leads to no
reward. The decision maker has been trained on the task, and wants to obtain the maximum possible
reward. Since the state is hidden, the decision maker must use one or more *observations* to estimate
the state. For example, the state could be one of two biased coins, one biased toward heads and
the other toward tails. On each trial, the experimenter picks one of these coins randomly and flips
it. The decision maker only sees the result, heads or tails, and must guess which coin has been
picked. If she guesses correctly, she gets a reward immediately. If she fails, she gets nothing. In
this context, *Accuracy* is defined as the number of correct guesses divided by the total number of
trials. In a single trial, if $A$ represents the action (or choice) of the decision maker, and $S$ and $Z$
denote the state and observation respectively, then Accuracy for the choice $s$ with observation $z$ is
the probability $P(A = a_s | S = s, Z = z)$ where $a_s$ represents the action of decision maker, i.e.
choosing $s$, and $s$ is the true state. This Accuracy can be measured by the experimenter. However,
from the decision maker's perspective, her chance of success in a trial is given by the probability
of $s$ being the correct state, given observation $z$: $P(S = s | Z = z)$. We call this probability the
decision maker's *belief*. After choosing an action, for example $a_s$, the *confidence* for this choice
is the probability: $P(S = s | A = a_s, Z = z)$. According to Bayes theorem:

$$P(A|S, Z)P(S|Z) = P(S|A, Z)P(A|Z). \tag{1}$$

As the goal of our decision maker is to maximize her reward, she picks the most probable
state. This means that on observing $z$ she picks $a_{s^*}$ where $s^*$ is the most probable state, i.e.
$s^* = \arg\max(P(S = s | Z = z))$. Therefore, $P(A|Z = z)$ is equal to 1 for $a_{s^*}$ and 0 for the
rest of the actions. As a result, accuracy is 1 for the most probable state and 0 for the rest. Also
$P(S|A, Z)$ is equal to $P(S|Z)$ for the most probable state. This means that, given observation $z$,
Accuracy is equal to the confidence on the most probable state. Also, this confidence is equal to the
belief of the most probable state. As confidence cannot be defined on actions not performed, one
could consider confidence on the most probable state only, implying that accuracy, confidence, and
belief are all equal given observation $z$:

$$\sum_s P(A = a_s | S = s, Z)P(S = s | Z) = P(S = s^* | A = a_{s^*}, Z) = P(S = s^* | Z).^1 \tag{2}$$

All of the above equalities, however, depend on the ability of the decision maker to compute $P(S|Z)$.
According to Bayes' theorem $P(S|Z) = P(Z|S)P(S)/P(Z) \quad (P(Z) \neq 0)$. If the decision maker
has the perfect observation model $P(Z|S)$, she could compute $P(S|Z)$ by estimating $P(S)$ and

values on those states.

$P(Z)$ beforehand by counting the total number of occurrences of each *state* without considering any observations, and the total number of occurrences of observation $z$, respectively. Therefore, accuracy and confidence are equal if the decision maker has the true model for observations. Sometimes, however, the decision maker does not even have access to $Z$. For example, in the motion discrimination task, if the data is binned based on difficulty (i.e., motion strength), the decision maker cannot estimate $P(S|\textit{difficulty})$ because she does not know the difficulty of each trial. As a result, accuracy and confidence are not equal.

In the general case, the decision maker can utilize multiple observations over time, and perform an action on each time step. For example, in the coin toss problem, the decision maker could request a flip multiple times to gather more information. If she requests a flip two times and then guesses the state to be the coin biased toward heads, her actions would be *Sample, Sample, Choose heads*. She also has two observations (likely to be two $Heads$). In the general case, the state of the environment can also change after each action.[2] In this case, the relationship between accuracy and confidence at time $t$ after a sequence (history $H_t$) of actions and observations $h_t = a_0, z_1, a_2, ..., z_{t-1}, a_{t-1}$, is:

$$P(A_t|S_t, H_t)P(S_t|H_t) = P(S_t|A_t, H_t)P(A_t|H_t). \tag{3}$$

With the same reasoning as above, accuracy and confidence are equal if and only if the decision maker has access to all the observations and has the true model of the task.

## 3  The POMDP Model

Partially Observable Markov Decision Processes (POMDPs) provide a mathematical framework for decision making under uncertainty in autonomous agents [8]. A POMDP is formally a tuple $(S, A, Z, T, O, R, \gamma)$ with the following description: $S$ is a finite set of states of the environment, $A$ is a finite set of possible actions, and $Z$ is a finite set of possible observations. $T$ is a transition function defined as $T : S \times S \times A \rightarrow [0, 1]$ which determines $P(s|s', a)$, the probability of going from a state $s'$ to another state $s$ after performing a particular action $a$. $O$ is an observation function defined as $O : Z \times A \times S \rightarrow [0, 1]$, which determines $P(z|a, s)$, the probability of observing $z$ after performing an action $a$ and ending in a particular state $s$. $R$ is the reward function, defined as $R : S \times A \rightarrow \mathbb{R}$, determining the reward received by performing an action in a particular state. $\gamma$ is the discount factor, which is always between 0 and 1, and determines how much rewards in the future are discounted compared to current rewards.

In a POMDP, the goal is to find a sequence of actions to maximize the expected discounted reward, $E_{s_t}[\sum_{t=0}^{\infty} \gamma^t R(s_t, a_t)]$. The states are not fully observable and the agent must rely on its observations to choose actions. At the time $t$, we have a history of actions and observations: $h_t = a_0, z_1, a_1, ..., z_{t-1}, a_{t-1}$. The *belief state* [1] at time $t$ is the posterior probability over states given this history and the prior probability $b_0$ over states: $b_t = P(s_t|h_t, b_0)$. As the system is Markovian, the belief state captures the sufficient statistics for the history of states and actions [19] and it is possible to obtain $b_{t+1}$ using only $b_t, a_t$ and $z_{t+1}$:

$$b_{t+1}(s) \propto O(s, a_t, z_{t+1}) \sum_{s'} T(s', s, a_t) b_t(s'). \tag{4}$$

Given this definition of belief, the goal of the agent is to find a sequence of actions to maximize the expected reward $\sum_{t=0}^{\infty} \gamma^t R(b_t, a_t)$. The actions are picked based on the belief state, and the resulting mapping, from belief states to actions, is called a policy (denoted by $\pi$), which is a probability distribution over actions $\pi(b_t) : P(A_t|b_t)$. The policy which maximizes $\sum_{t=0}^{\infty} \gamma^t R(b_t, a_t)$ is called the optimal policy, $\pi^*$. It can be shown that there is always a deterministic optimal policy, allowing the agent to always choose one action for each $b_t$ [20]. As a result, we may use a function $\pi^* : B \rightarrow A$ where $B$ is the space of all possible beliefs. There has been considerable progress in recent years in fast "POMDP-solvers" which find near-optimal policies for POMDPs [14, 17, 11].

### 3.1  Modeling Decision Making with POMDPs

Results from experiments and theoretical models indicate that in many perceptual decision making tasks, if the previous task state is revealed, the history beyond this state does not exert a noticeable

influence on decisions [2], suggesting that the Markov assumption and the notion of belief state is applicable to perceptual decision making. Additionally, since the POMDP model aims to maximize the expected reward, the problem of guessing the correct state in perceptual decision making can be converted to a reward maximization problem by simply setting the reward for the correct guess to 1 and the reward for all other actions to 0. The POMDP model also allows other costs in decision making to be taken into account, e.g., the cost of sampling, that the brain may utilize for metabolic or other evolutionarily-driven reasons. Finally, as there is only one correct hidden state in each trial, the policy is deterministic (choosing the most probable state), consistent with the POMDP model. All these facts mean that we could model the perceptual decision making with the POMDP framework. In the cases where all observations and the true environment model are available to the decision maker, the belief state in the POMDP is equal to both accuracy and confidence as discussed above. When some information is hidden from the decision maker, one can use a POMDP with that information to model accuracy and another POMDP without that information to model the confidence. If this hidden information is independent of time, we can model the difference with the initial belief state, $b_0$, i.e., we use two similar POMDPs to model accuracy and confidence but with different initial belief states. In the well-known motion discrimination experiment, it is common to bin the data based on the difficulty of the task. This difficulty is hidden to the decision maker and also independent of the time. As a result, the confidence can be calculated by the same POMDP that models accuracy but with different initial belief state. This case is discussed in the next section.

## 4    Experiments and Results

We investigate the applicability of the POMDP model in the context of two well-known tasks in perceptual decision making. The first is a fixed-duration motion discrimination task with a "sure option," presented in [13]. In this task, a movie of randomly moving dots is shown to a monkey for a fixed duration. After a delay period, the monkey must correctly choose the direction of motion (left or right) of the majority of the dots to obtain a reward. In half of the trials, a third choice also becomes available, the "sure option," which always leads to a reward, though the reward is less than the reward for guessing the direction correctly. Intuitively, if the monkey wants to maximize reward, it should go for the sure choice only when it is very uncertain about the direction of the dots. The second task is a reaction-time motion discrimination task in humans studied in [12]. In this task, the subject observes the random dots motion stimuli but must determine the direction of the motion (in this case, up or down) of the majority of the dots as fast and as accurately as possible (rather than observing for a fixed duration). In addition to their decision regarding direction, subjects indicated their confidence in their decision on a horizontal bar stimulus, where pointing nearer to the left end meant less confidence and nearer to the right end meant more confidence. In both tasks, the difficulty of the task is governed by a parameter known as "coherence'" (or "motion strength'"), defined as the percentage of dots moving in the same direction from frame to frame in a given trial. In the experiments, the coherence value for a given trial was chosen to be one of the following: $0.0\%, 3.2\%, 6.4\%, 12.8\%, 25.6\%, 51.2\%$.

### 4.1    Fixed Duration Task as a POMDP

The direction and the coherence of the moving dots comprise the states of the environment. In addition, the actions which are available to the subject are dependent on the stage of the trial, namely, random dots display, wait period, choosing the direction or the sure choice, or choosing only the direction. As a result, the stage of the trial is also a part of the state of the POMDP. As the transition between these stages are dependent on time, we incorporate discretized time as a part of the state. Considering the data, we define a new state for each constant $\Delta t$, each direction, each coherence, and each stage (when there is intersection between stages). We use dummy states to enforce the delay period of waiting and a terminal state, which indicates termination of the trial:

$$\text{S} = \{ \ (\textbf{d}\text{irection}, \textbf{c}\text{oherence}, \textbf{st}\text{age}, \textbf{t}\text{ime}), \text{ waiting states, terminal } \}$$

The actions are *Sample, Wait, Left, Right,* and *Sure*. The transition function models the passage of time and stages. The observation function models evidence accumulation only in the random dots display stage and with the action *Sample*. The observations received in each $\Delta t$ are governed by the number of dots moving in the same direction. We model the observations as normally distributed

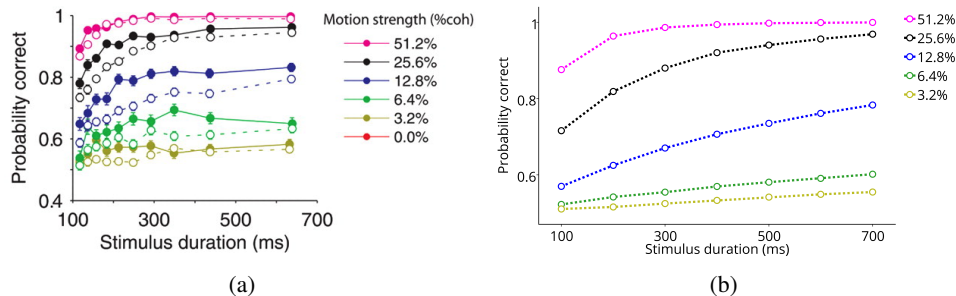

Figure 1: Experimental accuracy of the decision maker for each coherence is shown in (a). This plot is from [13]. The curves with empty circles and dashed lines are the trials where the sure option was not given to the subject. The curves with solid circles and solid lines are the trials where the sure option was shown, but waived by the decision maker. (b) shows the accuracy curves for the POMDP model fit to the experimental accuracy data from trials where the sure option was not given.

around a mean related to the coherence and the direction as follows:

$$O((d, c, display, t), Sample) = \mathcal{N}(\mu_{d,c}, \sigma_{d,c}).$$

The reward for choosing the correct direction is set to 1, and the other rewards were set relative to this reward. The sure option was set to a positive reward less than 1 while the cost of sampling and receiving a new observation was set to a negative "reward" value. To model the unavailability of some actions in some states, we set their resultant rewards to a large negative number to preclude the decision making agent from picking these actions. The discount factor models how much more immediate reward is worth relative to future rewards. In the fixed-duration task, the subject does not have the option of terminating the trial early to get reward sooner, and therefore we used a discount factor of 1 for this task.

### 4.2 Predicting the Confidence in the Fixed Duration Task

As mentioned before, confidence and accuracy are equal to each other when the same amount of information is available to the experimenter and the decision maker. Therefore, they can be modeled by the same POMDP. However, these two are not equal when we look at a specific coherence (difficulty), i.e. the data is binned based on coherence, because the coherence in each trial is not revealed to the decision maker. Figure 1a shows the accuracy vs. stimulus duration, binned based on coherence. The confidence is not equal to the accuracy in this plot. However, we could predict the decision maker's confidence only from accuracy data. This time, we use two POMDPs, one for the experimenter and one for the decision maker. At time $t$, $b_t$ of the experimenter's POMDP can be related to accuracy and $b_t$ of the decision maker's to confidence. These two POMDPs have the same model parameters, but different initial belief state. This is because the subject knows the environment model but does not have access to the coherence in each trial.

First, we find the set of parameters for the experimenter's POMDP to reproduce the same accuracy curves as in the experiment for each coherence. We only use data from the trials where the sure option is not given i.e. dashed curves in figure 1a. As the data is binned based on the coherence, and coherence is observable to the experimenter, the initial belief state of the experimenter's POMDP for coherence $c$ is as following: .5 for each of two possible initial states (at $time = 0$), and 0 for the rest. Fitting the POMDP to the accuracy data yields the mean and variance for each observation function and the cost for sampling. Figure 1b shows the accuracy curves based on the experimenter's POMDP.

Now, we could apply the parameters obtained from fitting accuracy data (the experimenter's POMDP) to the decision maker's POMDP to predict her confidence. The decision maker does not know the coherence in each single trial. Therefore, the initial belief state should be a uniform distribution over all initial states (all coherences, not only coherence of that trial). Also, neural data from experiments and post-decision wagering experiments suggest that the decision maker does not recognize the existence of a true zero coherence state (coherence = 0%) [13]. Therefore, the initial

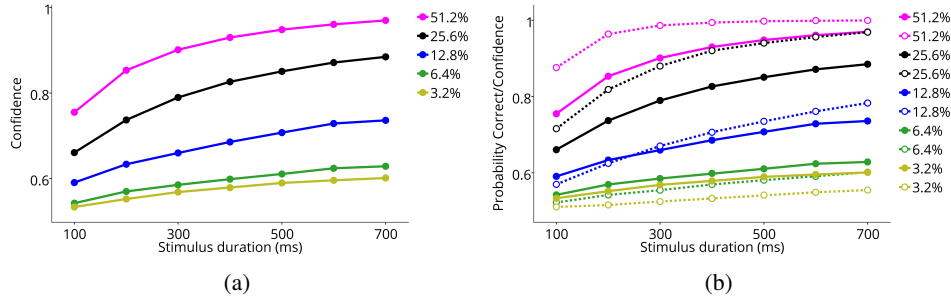

(a)    (b)

Figure 2: The confidence predicted by the POMDP fit to the observed accuracy data in the fixed-duration experiment is shown in (a). (b) shows accuracy and confidence in one plot, demonstrating that they are not equal for this task. Curves with solid lines show the confidence (same curves as (a)) and the ones with dashed lines show the accuracy (same as Figure 1b).

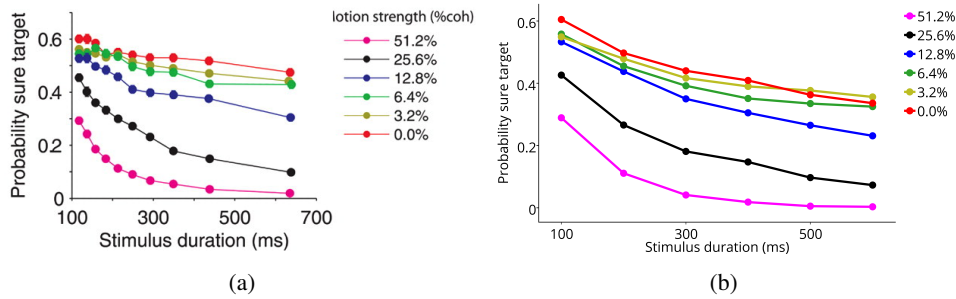

(a)    (b)

Figure 3: Experimental post-decision wagering results (plot (a)) and the wagering predicted by our model (plot (b)). Plot (a) is from [13].

probability of 0 coherence states is set to 0. Figure 2a shows the POMDP predictions regarding the subject's belief. Figure 2b confirms that the predicted confidence and accuracy are not equal.

To test our prediction about the confidence of the decision maker, we use experimental data from post-decision wagering in this experiment. If the reward for the sure option is $r_{sure}$ then the decision maker chooses it if and only if $b(right)r_{right} < r_{sure}$ and $b(left)r_{left} < r_{sure}$ where $b(direction)$ is the sum of the belief states of all states in that direction. Since $r_{sure}$ cannot be obtained from the fit to the accuracy data, we choose a value for $r_{sure}$ which makes the prediction of the confidence consistent with the wagering data shown in Figure 3a. We found that if $r_{sure}$ is approximately two-thirds the value of the reward for correct direction choice, the POMDP model's prediction matches experimental data (Figure 3b). A possible objection is that the free parameter of $r_{sure}$ was used to fit the data. Although $r_{sure}$ is needed to fit the exact probabilities, we found that any reasonable value for $r_{sure}$ generates the same trend of wagering. In general, the effect of $r_{sure}$ is to shift the plots vertically. The most important phenomena here is the relatively small gap between hard trials and easy trials in Figure 3b. Figure 4a shows what this wagering data would look like if the decision maker knew the coherence in each trial and confidence was equal to the accuracy. The difference between these two plots (figure 3b and figure 4a), and figure 2b which shows the confidence and the accuracy together confirm the POMDP model's ability to explain hard-easy effect [7], wherein the decision maker underestimates easy trials and has overconfidence in the hard ones.

Another way of testing the predictions about the confidence is to verify if the POMDP predicts the correct accuracy in the trials where the decision maker *waives* the sure option. Figure 4b shows that the results from the POMDP closely match the experimental data both in post-decision wagering and accuracy improvement. Our methods are presented in more detail in the supplementary document.

## 4.3 Reaction Time Task

The POMDP for the reaction time task is similar to the fixed duration task. The most important components of the state are again direction and coherence. We also need some dummy states for the

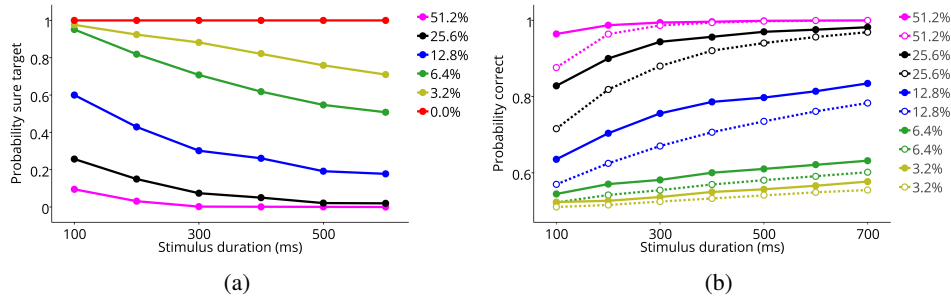

(a)                                              (b)

Figure 4: (a) shows what post-decision wagering would look like if the accuracy and the confidence were equal. (b) shows the accuracy predicted by the POMDP model in the trials where the sure option is shown but waived (solid lines), and also in the trials where it is not shown (dashed lines). For comparison, see experimental data in figure 1a.

waiting period between the decision command from the decision maker and reward delivery. However, the passage of stages and time are not modeled. The absence of time in the state representation does not mean that the time is not modeled in the framework. Tracking time is a very important component of any POMDP especially when the discount factor is less than one. The actions for this task are *Sample, Wait, Up* and *Down* (the latter two indicating choice for the direction of motion). The transition model and the observation model are similar to those for the Fixed duration task.

S = (**d**irection, **c**oherence), waiting states, terminal        $O((d,c), Sample) = \mathcal{N}(\mu_{d,c}, \sigma_{d,c})$

The reward for choosing the correct direction is $1$ and the reward for sampling is a small negative value adjusted to the reward of the correct choice. As the subject controls the termination of the trial, the discount factor is less than $1$. In this task, the subjects have been explicitly advised to terminate the task as soon as they discover the direction. Therefore, there is an incentive for the subject to terminate the trial sooner. While sampling cost is constant during the experiment, the discount factor makes the decision making strategy dependent on time. A discount factor less than $1$ means that as time passes, the effective value of the rewards decreases. Also, in a general reaction time task, the discount factor connects the trials to each other. While models usually assume each single trial is independent of the others, trials are actually dependent when the decision maker has control over trial termination. Specifically, the decision maker has a motivation to terminate each trial quickly to get the reward, and proceed to the next one. Moreover, when one is very uncertain about the outcome of a trial, it may be prudent to terminate the trial sooner with the expectation that the next trial may be easier.

### 4.4   Predicting the Confidence in the Reaction time Task

Like the fixed duration task, we want to predict the decision maker's confidence on a specific coherence. To achieve this, we use the same technique, i.e., having two POMDPs with the same model and different initial belief states. The control of the subject over the termination of the trial makes estimating the confidence more difficult in the reaction time task. As the subject decides based on her own belief, not accuracy, the relationship between the accuracy and the reaction time, binned based on difficulty is very noisy in comparison to the fixed duration task (the plots of this relationship are illustrated in the supplementary materials of [12]). Therefore we fit the experimenter's POMDP to two other plots, reaction time vs. motion strength (coherence), and accuracy vs. motion strength (coherence). The first subject (S1) of the original experiment was picked for this analysis because the behavior of this subject was consistent with the behavior of the majority of subjects [12]. Figures 5a and 5b show the experimental data from [12]. Figure 5c and 5d show the results from the POMDP model fit to experimental data. As in the previous task, the initial belief state of the POMDP for a coherence $c$ is $.5$ for each direction of $c$, and $0$ for the rest.

All the free parameters of the POMDP were extracted from this fit. Again, as in the fixed duration task, we assume that the decision maker knows the environment model, but does not know about the coherence of each trial and existence of $0\%$ coherence. Figure 6a shows the reported confidence from the experiments and figure 6b shows the prediction of our POMDP model for the belief of the

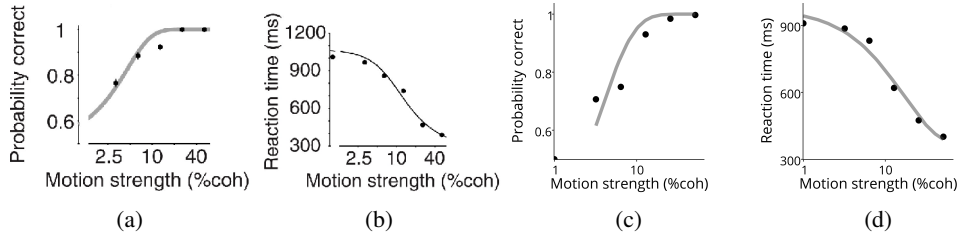

(a)     (b)     (c)     (d)

Figure 5: (a) and (b) show Accuracy vs. motion strength, and reaction time vs. motion strength plots from the reaction-time random dots experiments in [12]. (c) and (d) show the results from the POMDP model.

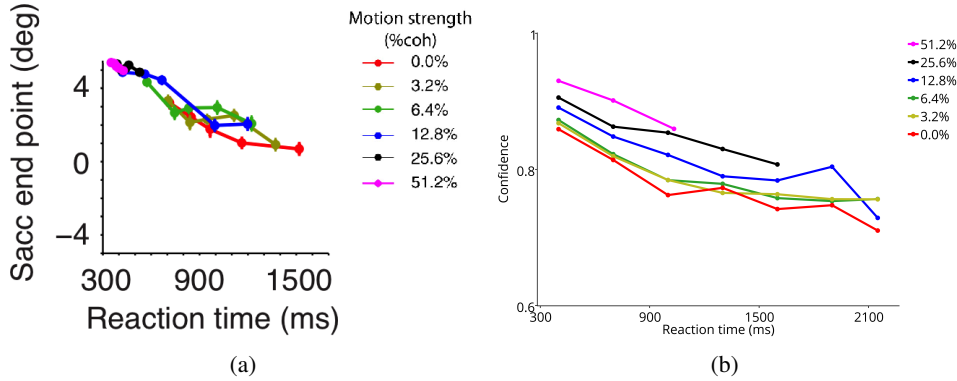

(a)     (b)

Figure 6: (a) illustrates the reported confidence by the human subject from [12]. (b) shows the predicted confidence by the POMDP model.

decision maker. Although this report is not in percentile and quantitative comparison is not possible, the general trends in these plots are similar. The two become almost identical if one maps the report bar to the probability range.

In both tasks, we assume that the decision maker has a nearly perfect model of the environment, apart from using 5 different coherences instead of 6 (the zero coherence state assumed not known). This assumption is not necessarily true. Although, the decision maker understands that the difficulty of the trials is not constant, she might not know the exact number of coherences. For example, she may divide trials into three categories: easy, normal, and hard for each direction. However, these differences do not significantly change the belief because the observations are generated by the true model, not the decision maker's model. We tested this hypothesis in our experiments. Although using using a separate decision maker's model makes the predictions closer to the real data, we used the true (experimenter's) model to avoid overfitting the data.

## 5   Conclusions

Our results present, to our knowledge, the first supporting evidence for the utility of a Bayesian reward optimization framework based on POMDPs for modeling confidence judgements in subjects engaged in perceptual decision making. We showed that the predictions of the POMDP model are consistent with results on decision confidence in both primate and human decision making tasks, encompassing fixed-duration and reaction-time paradigms. Unlike traditional descriptive models such as drift-diffusion or race models, the POMDP model is normative and is derived from Bayesian and reward optimization principles. Additionally, unlike the traditional models, it allows one to model optimal decision making *across* trials using the concept of a discount factor. Important directions for future research include leveraging the ability of the POMDP framework to model intra-trial probabilistic state transitions, and exploring predictions of the POMDP model for decision making experiments with more sophisticated reward/cost functions.

## Footnotes

[1]In the case that there are multiple states with maximum probability, Accuracy is the sum of the confidence

[2]In traditional perceptual decision making tasks such as the random dots task, the state does not usually change. However, our model is equally applicable to this situation.

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
