[Supplementary Material]



# Supplementary Document for:
# A Bayesian Framework for Modeling Confidence in Perceptual Decision Making

**Koosha Khalvati, Rajesh P. N. Rao**
Department of Computer Science and Engineering
University of Washington
Seattle, WA 98195
`{koosha, rao}@cs.washington.edu`

## Abstract

This is the supplementary document for "A Bayesian Framework for Modeling Confidence in Perceptual Decision making". In this document, we explain the process of fitting the experimenter's POMDP to the data and predicting the confidence by using decision maker's POMDP in detail.

## 1 The Model in Summary

There are two POMDPs: one for the experimenter and one for the subject. At time t, $b_t$ of the experimenter's POMDP can be related to accuracy and $b_t$ of the subject's to confidence. The experimenter has the perfect model of the environment. The subject has the same model except for awareness of the $0\%$ coherence state. The main difference however is not the model but the initial belief state ($b_0$). When the data is binned based on coherence, the experimenter knows the coherence in each single trial, but not the subject. As a result, $b_0$ of the experimenter is .5 in the two states with corresponding coherence (two directions), and 0 for the rest. $b_0$ of the subject is always a uniform distribution among all states but $0\%$ coherence (which has zero probability).

As the POMDPs share the same model parameters, to predict the confidence from accuracy, we first find the free parameters (observation function parameters and rewards) by fitting the experimenter's POMDP to the accuracy data. In other words, we search for parameters that fit the POMDP with the knowledge of the coherence in its initial belief state to the accuracy data. All accuracy data are extracted from the plots of the original papers. After finding the parameters, we apply them to the subject's POMDP which has the same model, but different initial belief state. The resultant $b_t$ in the subject's POMDP is her confidence at time $t$.

## 2 Fitting Process

The free parameters are: mean and variance of observation function for each coherence and direction, cost of sampling, reward of sure option in the fixed duration task, and the discount factor in the reaction time task. As the experiments are unbiased, we assume that the means are in the range of $[-1, 1]$ where $0.0\%$ coherence has a mean of 0 and $\mu_{left,c} = -\mu_{right,c}$. Also, for simplicity in the fitting process we assume that all variances are equal. Therefore, we need to find 6 parameters in total for the observation function: the mean of nonzero coherences only in one of the directions, and the variance which is the same for all coherences and directions.

We use a $k$-step look-ahead online search with sampling [3] as the solver for all POMDPs ( $k = 1$ for the fixed duration task and $k = 2$ for the reaction time task). Normally, a POMDP's result is evaluated by performing simulation many times and reporting the average discounted reward. On

each instance of the simulation, the initial true state is picked randomly from the initial belief state distribution. After that, at each step, the next observation is picked randomly from the observation distribution of the true state. Also, the next true state after each action is picked randomly from the transition function applied to the current true state. We do not need to be worried about the actions in our experiments as they are deterministic. As mentioned, the true state is picked from initial belief state because the assumption is that the agent has the perfect model of the environment, and also the true initial probability of the states. This assumption is true for the experimenter in our task. As a result, for experimenter's POMDP we do the same, i.e. picking the true state from the initial belief state of the POMDP [1]. However, the decision maker does not have the correct initial state. Therefore, in the simulation of the decision maker's POMDP, the true state is picked from the experimenter's initial belief state. Sampling from the observation probability distribution of the true state is the same for both POMDPs as both have the true model of the environment.

## 2.1  Parameter fitting in the fixed duration task

To find the free parameters, we fit the experimenter's POMDP to the data in figure 1a in the paper. We search for the 6 parameters of the observation function, and the cost of sampling. We use the data points in the trials where the sure option is not shown to the subject (dotted lines). As mentioned in the paper, finding $r_{sure}$ is not possible in this fitting. In fact, we intentionally refuse to use the data with the sure option (solid lines) to make the effect of decision maker's internal belief on her behavior as low as possible. We start from a guessed value for each parameter and then gradually change the values to get a closer fit. There are 5 different coherences, and we check the fitness at 7 data points for each coherence (each 100ms). For each data point at time $t$ and coherence $c$, the average belief of experimenter's POMDP at that time ($b_t$) should be equal to accuracy at that time. This average belief is calculated by running 1000 simulations of the POMDP for each $t$ and $c$.

After finding parameter values that give the closest fit, we predict the confidence by generating $b_t$ for the subject's POMDP (figure 2). In other words, the confidence at time $t$ is the average $b_t$ in the subject's POMDP at that time. This average is from 1000 simulations for each time $t$ and coherence $c$.

To predict the probability of choosing the sure option, we count the number of times where $b_t(right)r_{right} < r_{sure}$ and $b_t(left)r_{left} < r_{sure}$ out of 1000 trials, for each time $t$ and coherence $c$ in the subject's POMDP. As mentioned in the paper, the process of finding $r_{sure}$ is a part of this prediction. We find $r_{sure}$ equals 0.68. $b_t(direction)$ is the sum of all belief states in that direction in time $t$.

$$b_t(right) = \sum_c b_t((right, c)) \tag{1}$$

$$b_t(left) = \sum_c b_t((left, c)) \tag{2}$$

To show what the wagering plot would look like if confidence were equal to accuracy, we performed the same process described in the previous paragraph with experimenter's POMDP as well. In other words we count the number of times where $b_t(right)r_{right} < r_{sure}$ and $b_t(left)r_{left} < r_{sure}$ out of 1000 trials, for each time $t$ and coherence $c$ in the experimenter's POMDP. This plot is shown in figure 4a in the paper.

Another prediction is the performance of the decision maker when she waives the sure option (solid lines in figure 1a). For this, we run both POMDPs together (feeding the same observation at each step) and report accuracy when confidence is more than .68 for each time and coherence. In other words, we report $b_t$ of experimenter's POMDP when $b_t(right)r_{right} >= r_{sure}$ or $b_t(left)r_{left} >= r_{sure}$ [2]for $b_t$ in the subject's POMDP for each time $t$ and coherence $c$. The report which is shown in figure 4b in the paper is the average of 1000 simulations for each time and coherence.

### 2.2   Parameter fitting in the Reaction time task

This time there are 8 free parameters: 6 for the observation function and one each for the sampling cost and discount factor. To find these parameters, we start from guessed values and start to search by fitting:

- The average final $b_t$ (when action($a_t$)=up/down) of the experimenter's POMDP to average accuracy, for each coherence (figure 5a in the paper)
- The average time to choosing a direction ($t$ when $a_t$ = up/down) in the subject's POMDP to the average reaction time in the experiment, for each coherence (fig 5b in the paper).

In other words, we find the parameters based on two questions: 1) When does the subject stop observing and choose a direction? 2) With what belief does the subject make her choice? Also, as in the case of the accuracy plot when the subject waives the sure option in the fixed duration task, the same observation is fed to both of the POMDPs at each step.

Each average value is obtained from 1000 simulations. After finding the parameters, the confidence vs reaction-time plot is predicted by simulating the subject's POMDP and reporting the final $b_t$ (when $a_t$=up/down). Again, the result is from 1000 simulations for each coherence. In all of the POMDPs, a delay of $200ms$ is added to account for non-decision making processes [1]. In other words, $b_t$ is compared to accuracy or confidence at time $t + 200ms$.

To fit accuracy, both POMDPs are simulated simultaneously in the reaction time task simulations, until the decision maker ends the trial. However, this is true to some extent in the fixed duration task as well. Although the subject cannot "end" the trial, but she could stop sampling and wait till the end of the stimulus. However, we don't have to run both POMDPs in this particular fixed duration experiment. As figure 1a shows, the accuracy always improves with time. This means that the cost of sampling is so low that at least in the first $700ms$, the decision maker keeps sampling. Therefore, simulating the experimenter's POMDP alone is sufficient.

## Footnotes

[1]In fact, in our experiments, we pick only from one of the directions like the real experiments. Due to the symmetry in the model however, this does not change the resultant average reward, decision time or the probability of the dominant direction (and consequently the other direction).

[2]In the case of exact equality, the strategy of the decision maker is unpredictable. The probability of this case, however, is extremely low (practically zero).