[Reviews · NeurIPS 2015]

Submitted by Assigned_Reviewer_1

Major issues:

1. Too many technical details for simulations are missing-e.g., sure rewards, cost of sampling, discount factor.

2. The assumption that the decision maker knows the possible states (coherences) seems entirely unrealistic for the usual experimental scenario, where that information is not provided. As noted on line 256, coherence is observable to the experimenter, but not to the subject.

3. The paper stakes the claim that the POMDP model is better than a DDM, but no comparisons are actually made.

4. The logic of using two POMDPs (with and without the environment model) as models for accuracy and confidence was not clear to me.

5. The virtues of normativity for modeling human behaviour are not adequately defended here. Why would normativity be good in and of itself?

6. Does the model make any clear, testable predictions? I don't see any adduced here-only fits to the existing data.

Smaller bits.

-The figures were very hard to read. The lines were thin and dots tiny. The modeling plots should strive to be like the data plots (e.g., 1A, 3A, 6A). Also, the legend should appear with each plot. - What temporal dynamics cannot be captured by a DDM (implied on line 55). -line 79, trial (not trail). also in caption to Fig 4. -line 140. goal is maximizing expected discounted reward (not total) -line 208. actions ... are -Fig 5 was entirely unreadable. -The data in Fig 6 (sacc end point) do not match the supposed dependent variable (confidence).
Summary: This paper presents a POMDP model of confidence judgments in decision making. The simulations, however, are vaguely described and the case for how this model compares with existing approaches is not strong.

Submitted by Assigned_Reviewer_2

The authors' model confidence data from two experiments (conducted by others and previously published in the scientific literature) using a POMDP. In both experiments, subjects saw a random-dot kinematogram on each trial and made a binary choice about the dominant motion direction. The first experiment used monkeys as subjects and stimuli had a fixed duration. The second experiment used people as subjects and stimuli continued until a subject made a response. The paper reports that the POMDP model does a good job of fitting the experimental data, both the accuracy data and the confidence data.

I should start by stating that I know a bit about perceptual decision making and a (small) bit about POMDPs but I am unfamiliar with the application of POMDPs to perceptual decision making. The paper reports that this is the first time that a POMDP has been used to model confidence data in a perceptual decision making task. I'll take the authors' word on this.

Taken as a whole, I have a positive opinion of this manuscript. I very much like the problem studied here (modeling confidence data in perceptual decision making) and I like the idea of applying a POMDP to this data. The fit to the experimental data is qualitatively good (though quantitatively mediocre).

A significant shortcoming of this paper is the lack of a literature review section providing context for the work provided here. Other researchers have previously applied POMDPs to perceptual decision making. Did they only model accuracy data? Did they model reaction time data too? How does the POMDP studied here relate to the POMDPs presented in these earlier papers? Is it largely the same or is it substantially different? If the latter , how (and why) is it different? If the current paper is the first to apply POMDPs to confidence data in a perceptual decision making task, how did researchers model confidence data previously? Without answers to these questions, there is no way to evaluate the novelty and significance of the contribution made by the current paper.
Summary: I like the problem studied here, and I like the application of POMDPs to confidence data in the context of perceptual decision tasks. However, the paper does not include an adequate literature review section and, thus, it is hard to know the extent to which the reported research is a novel and significant contribution above and beyond what already exists in the literature.

Author Feedback
Author rebuttal: We wish to thank all the reviewers for their insightful comments. Any new explanations presented here will be added to the paper as well. We will also include a supplementary doc containing more details of the model and simulations. These details will be explained here in summary. Also, the plots will be improved.

Model (Reviewer 1,3)
There are two POMDPs: one for the experimenter and one for the subject. At time t, b_t of the experimenter's POMDP can be related to accuracy and b_t of the subject's to confidence. The experimenter has the perfect model of the environment. The subject has the same model except for awareness of the 0% coherence state. The main difference however, is not the model but the initial belief state (b_0). When the data is binned based on the coherence, the experimenter knows the coherence in each single trial, but not the subject. As a result, b_0 of the experimenter is .5 in the two states with corresponding coherence (two directions), and 0 for the rest. b_0 of the subject is always a uniform distribution among all states.

Simulation (Reviewer 1,3,4)
Fixed duration: To find free parameters we fit the experimenter's POMDP to fig 1a. We start from a guessed value for each parameter and then gradually change the values to get a closer fit. There are 5 different coherences, and 7 data points for each coherence (for each 100ms). For each data point at time t and coherence c, the average belief of experimenter's POMDP at that time (b_t) should be equal to accuracy at that time. This average belief is calculated by running 1000 simulations of POMDP for each t and c. After finding parameter values that give the closest fit, we predict confidence and wagering by generating b_t for the subject's POMDP (fig 2 & 3). Again the average b_t is from 1000 simulations. Parameters found: r_sure = 0.68, r_sample = -10^-5.
Reaction time: To find free parameters, we start from guessed values and start to search by fitting: 1) The average final b_t (when action(a_t)=up/down) of experimenter's POMDP to average accuracy, for each coherence (fig 5a); 2) The average time to choosing a direction (t when a_t = up/down) in the subject's POMDP to average reaction time in the experiment, for each coherence (fig 5b). Each average value is obtained from 1000 simulations. After finding the parameters, the confidence v.s reaction-time plot is predicted by simulating subject's POMDP and using the final b_t (when a_t=up/down) (average of 1000 simulations for each coherence). Parameters found: gamma = 0.97, r_sample = -10^-5.

Novelty in POMDP (Reviewer 2)
This is the first time that confidence has been modelled with POMDPs. Other POMDP papers only considered accuracy. The focus of [18] (cited in submission) is on network implementation of the POMDP rather than prediction in experiments. [5] and [6] test their predictions on experiments but only on the reaction time task (not fixed duration) and only on accuracy vs. coherence and reaction time vs. coherence (not accuracy vs. reaction time). There is no discount factor in their model and cost of actions remains constant throughout the trial. Our confidence model and our discount factor explains for the first time certain important attributes of task such the absence of reaction time vs. accuracy analysis in previous work and why confidence decreases with time (lines 366-370 & fig 6). The material on mathematical soundness of the POMDP approach (sections 2 & 3.1) is new as well.

DDM (Reviewer 1)
As mentioned in the introduction, a separate optimizer is needed for the DDM to include costs. Also, not all kinds of uncertainties can be incorporated in the DDM. For example, in modeling confidence, there is an uncertainty over coherence, which is the drift rate in the DDM. In the classical DDM, there is no uncertainty over the drift rate. One could add functions to DDM to cover these situations, but this confirms that DDM is not a unifying and universal model for decision-making and needs to be modified in an ad-hoc manner for different tasks.

Normativity (Reviewer 1)
Normative models are important because they allow us to compare biological strategies to how an optimal agent 'should' behave under some explicit notion of what the agent is trying to optimize.

Qualitative results (Reviewer 7,1)
While exact quantitative fits can provide interesting insights, we have opted for qualitative fits for the following reasons. First, using extra parameters, e.g., mapping between saccade endpoint and probability (fig 6, lines 407-409) or assuming an imperfect model for the subject (lines 411-417) can lead to overfitting and hurt the generality of the framework. Also, the main contribution of this paper is presenting a new normative framework for confidence based on POMDPs and demonstrating its mathematical soundness. The data presented has been previously published and not new - the emphasis is on confidence prediction within the context of the POMDP framework.